# Restricted Access to Working Memory Does Not Prevent Cumulative Score Improvement in a Cultural Evolution Task

**DOI:** 10.3390/e24030325

**Published:** 2022-02-24

**Authors:** Juliet Dunstone, Mark Atkinson, Elizabeth Renner, Christine A. Caldwell

**Affiliations:** 1Psychology, University of Birmingham, Birmingham B15 2TT, UK; 2School of Management, University of St Andrews, St Andrews KY16 9RJ, UK; markdatkinson64@gmail.com; 3Psychology, University of Durham, Durham DH1 3LE, UK; elizabeth.renner@durham.ac.uk; 4Psychology, University of Stirling, Stirling FK9 4LA, UK; c.a.caldwell@stir.ac.uk

**Keywords:** cumulative culture, cultural evolution, working memory, metacognition, dual-task

## Abstract

Some theories propose that human cumulative culture is dependent on explicit, system-2, metacognitive processes. To test this, we investigated whether access to working memory is required for cumulative cultural evolution. We restricted access to adults’ working-memory (WM) via a dual-task paradigm, to assess whether this reduced performance in a cultural evolution task, and a metacognitive monitoring task. In total, 247 participants completed either a grid search task or a metacognitive monitoring task in conjunction with a WM task and a matched control. Participants’ behaviour in the grid search task was then used to simulate the outcome of iterating the task over multiple generations. Participants in the grid search task scored higher after observing higher-scoring examples, but could only beat the scores of low-scoring example trials. Scores did not differ significantly between the control and WM distractor blocks, although more errors were made when under WM load. The simulation showed similar levels of cumulative score improvement across conditions. However, scores plateaued without reaching the maximum. Metacognitive efficiency was low in both blocks, with no indication of dual-task interference. Overall, we found that taxing working-memory resources did not prevent cumulative score improvement on this task, but impeded it slightly relative to a control distractor task. However, we found no evidence that the dual-task manipulation impacted participants’ ability to use explicit metacognition. Although we found minimal evidence in support of the explicit metacognition theory of cumulative culture, our results provide valuable insights into empirical approaches that could be used to further test predictions arising from this account.

## 1. Introduction

Cumulative cultural evolution (CCE) is the process by which cultural traits are transmitted through generations of cultural agents and, crucially, amended by successive generations to become more effective, efficient or beneficial for their users [1,2]. This process has been dubbed ‘the ratchet effect’ due to the unidirectional progress of improvement [3] and is generally considered to be a process that is unique to, or at least qualitatively distinctive in, humans [2,4,5]. Much research within cultural evolution therefore aims to identify the human-unique cognitive capacities that enable this gradual accumulation over generations.

A current theory argues that explicit, system-2 processes (discussed in detail below) which enable *strategic* learning from others, rather than a capacity for social learning and/or imitation, may explain the cumulative improvement of traits over generations [6,7]. This theory has been dubbed the Explicitly Metacognitive Cumulative Culture hypothesis (EMCC; see [6]). It posits that, by the time they reach adulthood, typically developing modern humans have an explicit awareness of which information is beneficial, and should therefore be retained, and which information can be discarded. This allows for not only the retention and transmission of cultural traditions, but their cumulative evolution over time. Such an awareness is believed to require a metacognitive understanding of one’s own knowledge and skill level, and an explicit awareness of the social learning strategy [8] to be applied in a particular context.

### 1.1. Measuring Cumulative Cultural Evolution

Demonstrating cumulative cultural evolution in the lab has traditionally been carried out using transmission chain methods (see [9,10]). However, this method is very labour intensive due to the large number of participants required, especially if comparing across multiple experimental conditions (for example, Caldwell & Millen (2009) [11] tested 700 participants). If wanting to make developmental or inter-species comparisons the large sample size required may present methodological barriers that are hard to overcome (although see [12,13,14]).

Caldwell et al. [15] proposed an alternative method of assessing the capacity for cumulative cultural evolution by testing the propensity of individuals to improve upon information they are given at any one time. Caldwell et al. called this the individual’s *potential for ratcheting* (PFR). If presented with various pieces of information reflecting varying levels of success, a participant that was able to perform better (e.g., score more points) after observing a better (e.g., higher scoring) demonstration would be displaying some PFR. If they were consistently able to outperform the demonstration, at multiple levels of success, this may reflect a level of PFR that was more analogous with cumulative cultural evolution. Being exposed to demonstrations that score higher than would be expected by chance or random exploration is the experimental equivalent of being exposed to a trait or tool that has been modified by a conspecific to be more useful than in its natural state in the environment: for example, exposure to knapped stone tools as opposed to unmodified rocks. This paradigm has been used to test the potential for ratcheting in tufted capuchins [16] and young children [17].

### 1.2. Measuring EMCC

To assess the validity of the EMCC hypothesis, we need to be able to test whether an observed behaviour (ratcheting) relies on explicit processes, specifically, the system-2 processes as defined by Evans and Stanovich [18]. The defining element of Evans and Stanovich’s conceptualisation of system-2 is that it relies on working memory. Whether PFR relies upon system-2 processes can therefore be investigated by comparing performance in a PFR task with and without access to working memory resources. Wilks et al. (2021) [17] found that PFR in young children was evident only when there was no memory requirement to the task. Task interference from memory demands was also found in a search task that required using a simple selective social learning strategy [19].

Restricting access to working memory resources via a dual-task paradigm has been argued to impede metacognitive responding in an uncertainty monitoring task [20] and a confidence rating task [21]. Following this line of reasoning, restricting access to working memory via a dual-task paradigm should therefore be a viable method to restrict usage of metacognitive processing. If trying to assess the involvement of explicit metacognitive processes in PFR, a dual-task paradigm which taxes working memory can therefore be used as a proxy to restrict metacognition directly.

The following study aimed to experimentally test the EMCC hypothesis by restricting access to participants’ working memory resources while they completed a task that evaluated their PFR. Participant scores were then used to model the expected outcome if the same task was iterated over many generations. We predicted that, under a working memory load, participants would use social information less efficiently due to a reduction in the information-processing resources available to a participant; the two concurrent tasks would be required to share the available ‘channel capacity’ [22]. We therefore predicted that the additional working memory load would result in a reduced PFR in individuals and less ratcheting over simulated generations.

Participants completed a grid search task under dual-task conditions that applied an additional working memory load. Participants were further split by cue type: half of the participants completed the task with visible cues in which squares revealed during the information trial were shown during each trial, and half with transient cues in which the reward values of squares were shown briefly and then disappeared (i.e., introducing a memory requirement to the grid search task itself). This was included to add an additional, ecologically valid (although see below), memory load to the copying strategy requirement. In many ‘real world’ copying scenarios the behaviour to be copied will not remain visible while it is being copied or may not leave an obvious physical trace of having been completed, such as if a resource being foraged does not show signs of being depleted after a foraging event (e.g., water sources).

The study also aimed to investigate the impact of the same concurrent working memory load on metacognitive efficiency (defined as metacognitive sensitivity divided by overall task sensitivity—see results section). If metacognitive efficiency is significantly negatively impacted by the restriction of working memory access, this would lend support to the EMCC, and potentially shed light on the mechanisms involved in any link found between PFR and working memory load. However, if PFR is reduced under working memory load but metacognition is not this may implicate the role of system-2, but not necessarily metacognitive processes in cumulative cultural evolution.

The tasks presented below are not aiming to have very high ecological validity; due to the nature of a dual-task paradigm needing to be tightly constrained in order to rule out confounding task interactions, the tasks are deliberately simple and not very challenging for participants. The value of this simplicity is that it ensures we can test a very specific hypothesis such as the EMCC that might not be practical or produce clear results with a more complex or ‘realistic’ cultural evolution task. Once the feasibility of using the methods outlined below to test the EMCC has been established, progressing to more complex or challenging tasks would be a valuable addition to this field of research.

## 2. Methods

### 2.1. Design

Participants were randomly allocated to one of two ‘main task’ conditions—grid search and metacognition. Participants assigned to the grid search task were then randomly assigned to one of two conditions (see *grid search task* below). A between-subjects design was used in order to keep the participation period short enough for testing in public. We aimed for a total participation time of around 15 min per participant to avoid task fatigue, low uptake or high drop-out rates (the latter two due to potential participants not wanting to take a long time out of a paid-entry science centre visit to participate). Each participant in both tasks was shown two blocks of 25 trials per block. The first trial from each block was a practise trial and was removed from analyses.

### 2.2. Equipment

Participants completed all the tasks on touchscreen Lenovo Yoga laptops. All interaction between the participant and the task was via the touchscreen. No keyboard or mouse were provided. Participants wore Goji over-ear active noise-cancelling headphones. The task was written in PsychoPy 3.0.3 and run in PsychoPy 1.84.2 [23]. Code to run the tasks is provided via the OSF (see Data Availability Statement).

### 2.3. Procedure

For both tasks, participants initially read a series of on-screen instructions. They were then asked 4 understanding check questions to ensure the instructions had been taken in as intended. If any of these questions were answered incorrectly the correct answer was displayed again on screen. In line with the pre-registration document (https://osf.io/w4zyj (accessed on 5 December 2021)) all participants scored 50% or higher in the understanding check. Participants were given an opportunity to ask any questions before beginning the task.

At one testing location (Centre for Life only, see below), after completing both blocks of the task participants were invited to write their scores up to be placed on a daily scoreboard that was visible to all participants and visitors to the centre. This was voluntary and participants were not required to provide their names.

#### 2.3.1. Grid Search Task

Each trial of the grid search task consisted of an information trial in which the participant was shown 5 selections on a 5 × 5 grid of white squares. These appeared automatically and were displayed for 1 s before the participant was required to make their own selections, with the aim being to find rewarded squares in the grid. Participants were randomly assigned to participate with either transient cues or visible cues (between subjects):Transient cues (GST): the information trial disappeared before the participant was required to make their own selection from the grid.Visible cues (GSV): the information trial remained visible (but with muted colours) while the participant was required to make their own selection from the grid.

The selections in the information trial showed between 0–5 rewards (six different levels of success) presented as green (rewarded) and red (unrewarded) squares. There were an equal number of trials with each number of rewards presented across the task, presented in a fully random order. Half a point was available for each correct selection made and there was a total of 5 green squares located randomly in each grid, so each grid was worth up to 2.5 points. Immediate feedback was given to participants about their correctness, with the selected square changing colour and an audio cue playing to indicate either a rewarded or unrewarded selection: rewarded selections made by the participant turned the grid square dark green and a ‘ping’ sound was played, unrewarded selections turned the square red and produced a ‘pop’ sound.

See Figure 1 for an example of a trial.

This task is similar in design to those used with capuchin monkeys [16] and children [17,24]. As this task explicitly labels the useful information that should be copied, the challenge is not simply to identify this information but to apply it efficiently in order to use a selective strategy—something which is often not implemented by the different study populations cited above.

#### 2.3.2. PFR Task Entropy

There are 53,130 (choose 5 from 25; $\left(\begin{array}{c} 25\\ 5 \end{array}\right)$) possible unique configurations of the five rewards within the grid, demonstrating that the task is not trivially easy as finding all five green squares without using the information provided in the information trial would be extremely unlikely. These unique configurations would lead to an entropy of 15.7 bits for the grid task, if no information trial was provided (log_2_$\left(\begin{array}{c} 25\\ 5 \end{array}\right)$) [25]. The information trial reduces the entropy by increasing the amount of known information about the grid and therefore reducing the number of places required for a participant to look for the solution. The amount of information provided by the information trial varies as a function of *χ*, where *χ* is the number of unrewarded (red) squares shown in the information trial:$$\log_{2}\left(\begin{array}{c} 25\\ 5 \end{array}\right)\;-\;\log_{2}\left(\begin{array}{c} 20\\ \chi \end{array}\right)$$

For example, the amount of information provided by the examples shown in Figure 1 is log_2_$\left(\begin{array}{c} 25\\ 5 \end{array}\right)$ − log_2_$\left(\begin{array}{c} 20\\ 2 \end{array}\right)$ = 8.1 bits.

#### 2.3.3. Metacognition Task

On each trial two patches of static dots were presented on screen and participants were asked to rate which patch had a higher density by selecting a point on a scale under their selected patch. The patches were randomly generated on each trial. The target patch was always 8% more dense than the foil patch, which would be generated with a dot density between 600 and 800 dots per patch. Whether the target appeared on the left or right of the screen was randomised for each trial, so there was an approximately equal split of targets on the left and right. This difficulty level aimed to achieve a discrimination accuracy of around 72% (based on a previous task which used 7 different difficulty levels and found 72% accuracy at the 8% level: [19]). A single difficulty level rather than a range of levels or titrated difficulty based on participant accuracy was used based on findings from [26] that suggest staircase procedures can inflate estimates of metacognitive efficiency.

The scale under each patch ran from 0–100 (0 in the centre, 100 at the far left/right), with participants instructed to rate their confidence along this scale. Verbal markers were given at points 100, 50 and zero on each scale saying, “definitely left/right”, “50% left/right” and “Guessing left/right”, respectively. The further along the scale the selection was made, the surer the participant was that the selected patch was correct. Participants could tap anywhere on the scale and this would automatically be rounded to the nearest 10. The confidence rating would then be converted to a decimal value (for example 40% confidence = 0.4).

After participants had made their rating, the scale with the selected confidence rating would remain visible and the participant was required to tap a statement at the top of the screen that asked them to confirm they were selecting a specific side with a specific level of confidence.

Points on each trial were awarded between 0–1 for the confidence ratings, scored using a strictly proper scoring rule: points = 1 − (accuracy–confidence)^2^. This means points were not awarded solely for correctness, but for the combination of accuracy and confidence. High scores were awarded for high confidence ratings given to correct responses and to low confidence ratings given to incorrect responses. Low scores were awarded to correct answers rated with low confidence and incorrect answers rated with high confidence. Participants were informed in the instructions that responses would be scored based on a combination of accuracy and confidence and told they could achieve the highest scores by rating their confidence accurately. Immediate feedback was given to participants about their score, and an audio cue indicated whether the selection was correct (‘ping’) or incorrect (‘pop’).

See Figure 1 for an example of a trial.

#### 2.3.4. Distractor Tasks

Each trial of both the grid search task and the metacognition task (herein referred to as ‘main tasks’) was sandwiched with a trial of a concurrent working memory distractor task, or a control distractor task.

#### 2.3.5. Working Memory Distractor Task

Before the onset of the main task trial, two single-digit numbers were presented on screen, one in a much larger font size than the other. The participant was instructed to remember both the size of the text and the value of the numbers. The numbers were then masked, and the trial of the main task commenced. After the trial of the main task was completed a memory probe question was asked, with participants required to recall either the large or small font size, or the large or small number value. Participants were asked to tap the side of the screen that contained the correct value from the start of the trial.

#### 2.3.6. Control Distractor Task

The control task followed the same task structure, but before a main task trial, instead of being shown two numbers to remember participants were shown two fixation crosses. After the main task trial, instead of a memory probe question two numbers were presented and a simple arithmetic question asking which of the two numbers presented was either larger or smaller was asked. Participants were asked to tap the correct answer.

All participants completed a block of trials with both the working memory and control distractor tasks, counterbalanced for block order.

For both the working memory and control block, 2 points were awarded for a correct response to the distractor task, and 2 points were lost for an incorrect response. This feedback was immediate via a visual score update and an audio cue (‘ping’ for correct, and a gameshow style ‘buzz’ for incorrect). A running total of the participant’s score was visible throughout the game.

Each trial of each main task had a 5 s time limit, after which the trial would timeout and move onto the next trial, skipping any remaining part of that trial. For example, if a participant had only selected two of their five selections in the grid search task within the time limit the trial would end and the memory probe question would not be asked. For trials that timed-out, only the points accrued in the first 5 s would be awarded. The exception to this was if during the metacognition task a participant had made a confidence rating but did not tap to confirm their selection within the 5 s. In these instances, the trial would continue as normal after a visual prompt that the confirmation had been missed and the working memory trial would be presented. Time limits were included firstly to ensure that participants were using their immediate working memory to recall the numbers at the end of each trial, and secondly to ensure the overall running time of the study remained short (see *design* above).

### 2.4. Participants

Data were collected both in a public science centre and a university lab. All participants gave written consent to take part. Ethical approval for the study was granted by the University of Stirling General University Ethics Panel (GUEP 600-602).

#### 2.4.1. Public Sample

All participants in the public data collection period were tested at the International Centre for Life in Newcastle-upon-Tyne, UK. Participants were not financially compensated for their time and participation was voluntary. All participants were informed they could stop participating at any time. One hundred and seventy four participants (79:95 male:female, mean age = 39.9, sd = 16.17) took part, split across the two main tasks (116 in the grid search task and 58 in the metacognition task). One participant took part in both tasks. A sample size of 40 participants per condition (for a total of 120 participants) were pre-registered (https://osf.io/w4zyj (accessed on 5 December 2021)). This was exceeded as the sample was reached earlier than expected, but the full data collection period was still completed out of courtesy to the science centre and on the basis that a larger sample could only improve the reliability of the results. The analysis presented in the results section uses the full dataset, as preliminary analysis showed both the full and reduced data sets produced models with the same significant effects.

Eleven participants were excluded in line with the exclusion criteria stated in the pre-registration document, for either choosing to leave before the task was completed or due to technical problems that occurred while they were taking part. A further four participants were excluded based on BPS guidelines for giving informed consent: one did not check the boxes to confirm consent was given, two did not put names on their consent forms and one participant was later discovered to have learning difficulties which may have meant that they were not capable of giving informed consent.

#### 2.4.2. Lab Sample

A large number (nine of fifty-five) of the public sample participants in one of the grid search task conditions always copied the hint exactly and never explored any of the other locations in the grid (see *grid search task* above). In case this strategy related to task understanding, the task instructions were therefore updated slightly before running further participants using the grid search task again in the lab (an updated pre-registration document was made https://osf.io/59pbf (accessed on 5 December 2021)). The metacognition task was not repeated in the lab.

Participants were recruited at the University of Stirling and took part in exchange for research participation tokens which were required for course completion. All participation was voluntary and participants were informed that they could stop testing at any time. Ninety participants (13:73:1 male: female: non-binary, mean age = 21.2, sd = 5.06) took part. These were split evenly across two testing conditions (45 in each). Data from one participant were removed from each condition: one as they chose to withdraw their data after participating and one as they skipped almost 40% of trials in one block of testing. All exclusions were in line with the pre-registration document, although the registered sample size was slightly exceeded due to the nature of the study sign-up system allowing for some overbooking.

Overall, the number of participants per condition is given in Table 1.

## 3. Results

### 3.1. Grid Search Task

Analysis was carried out on the combined sample from the Centre for Life and the lab, as the testing location was shown to have no significant impact on scores, outperformance or strategy use in the grid search task (*p* = 0.151, *p* = 0.116 and *p* = 0.324, respectively). All analysis below is for all test trials that were completed without a timeout (for a maximum of 24 per block), with the first practise trial removed for every participant. For all models, the *p*-values were estimated from the resultant t-statistics with degrees of freedom being the number of observations minus the number of fixed parameters in the model [27]. All models were significantly better than their null equivalent. The baseline for all the models presented below is the control block with visible cues, testing at the Centre for Life.

### 3.2. Overall Score

The overall score on each trial (see Figure 2) was analysed using a linear mixed effects model with fixed effects of block condition (control or working memory, herein WM), cue type (transient or visible), number of rewards in the information trial (rewards or ‘hits’) and testing location (Centre for Life or lab), and the interactions between block condition, cue type and rewards. Rewards were included as a random slope and participant ID as a random effect. Accuracy in the concurrent distractor task trial was not included as a random effect as this produced a singular fit. Overall score was significantly higher with the transient cue type (b = −0.146, SE = 0.056, t (8995) = −2.61, *p* = 0.009) and on trials where the information trial showed more rewards (b = 0.728, SE = 0.014, t (8995) = 50.6, *p* < 0.001). There were no significant effects of block condition (although this approached significance: b = −0.058, SE = 0.034, t (8995) = −1.69, *p* = 0.092) or testing location (b = −0.055, SE = 0.039, t (8995) = 1.44, *p* = 0.151) and no significant interactions (b ≤ |0.075|, SE ≤ 0.048, t (8995) ≤ |1.57|, *p* ≥ 0.117).

#### 3.2.1. Outperformance of the Information Trial

A measure of outperformance was calculated for each trial (see Figure 3), which quantified each participant’s potential to outperform the information trial on any given trial. This measure was included to assess not just whether participants scored more highly after observing better information, but whether they could display *potential for ratcheting* analogous to cumulative culture, by improving upon the information they have seen. This is calculated by subtracting the information trial score from the expected score on each trial. The expected score was calculated by assigning points based on strategy use: repeating a ‘hit’ gains half a point in the same manner as during game play, repeating the selection of a square shown to be unrewarded (a ‘miss’) gains 0 points, and exploration of the grid scores points proportionally to the likelihood of finding a hit in the remaining 20 unexplored grid squares. This score takes into account that a participant may be using a correct strategy and still not find rewarded squares in the grid, due to the element of chance required to make successful grid explorations. Take, for example, a trial in which the information trial showed 3 rewards. There would be 2 rewards, or ‘hits’ left to find in the grid out of the 20 remaining unexplored squares. If the participant repeated 3 ‘hits’, repeated 1 ‘miss’ and made 1 grid exploration, their expected score for that trial would be (3 × 0.5) + (1 × 0) + (1 × (0.5 × (2/20))) = 1.5 + 0 + 0.05 = 1.55. The information trial score for that trial would be 1.5, so the outperformance score would be 0.05.

Outperformance on each trial was analysed using a linear mixed effects model with fixed effects of block condition (control or WM), cue type (transient or visible), number of rewards in the information trial (rewards) and testing location (Centre for Life or lab), and the interactions between block condition, cue type and rewards. Rewards were included as a random slope and participant ID as a random effect. Participants were significantly less able to outperform the information trial when taking part with transient cues (b = −0.050, SE = 0.022, t (8995) = −2.29, *p* = 0.022) and significantly less able to outperform the information trial when it showed more rewards (b = −0.129, SE = 0.006, t (8995) = −20.4, *p* < 0.001), and there was a significant interaction between block condition and number of rewards (b = −0.009, SE = 0.003, t(8995) = −2.62, *p* = 0.009). Post hoc analysis using the emtrends function in R indicates that this interaction is due to significantly less decline in outperformance in the control block as rewards increase, as compared to the working memory block (b = 0.007, SE = 0.002, z = 3.14, *p* = 0.002). The effects of block condition (b = 0.008, SE = 0.010, t (8995) = 0.848, *p* = 0.396) and testing location (b = 0.027, SE = 0.017, t (8995) = 1.57, *p* = 0.116) were not significant, nor were the interactions between block and cue type, cue type and rewards, or between block, cue type and rewards (b ≤ |0.005|, SE ≤ 0.014, t (8995) ≤ |0.621|, *p* ≥ 0.535).

#### 3.2.2. Strategy Use

Correct strategy use (repeat rewarded selections and avoid unrewarded selections, ranging from 0 to5 on each trial) was analysed using a linear mixed effects model with fixed effects of block condition (control or WM), cue type (transient or visible), number of rewards in the information trial (rewards) and testing location (Centre for Life or lab), and the interactions between block condition, cue type and rewards. Rewards were included as a random slope and participant ID as a random effect. The WM block had significantly higher correct strategy use than the control block (b = 0.080, SE = 0.027, t (8995) = 2.93, *p* = 0.003), and the transient cue type showed significantly lower correct strategy use than the visible cue type (b = −0.708, SE = 0.189, t (8995) = −3.75, *p* < 0.001). There was a significant interaction between block condition and rewards (b = −0.030, SE = 0.009, t (8995) = −3.35, *p* < 0.001) and between cue condition and rewards (b = 0.129, SE = 0.041, t (8995) = 3.17, *p* = 0.002).

Post hoc testing using the emtrends function in R shows that, although correct strategy use in the WM block is higher than in the control block when rewards are low, it increases less steeply with increasing reward number in the WM block compared with the control block (b = 0.024, SE = 0.006, z = 3.80, *p* < 0.001). Strategy use increased significantly more with increased rewards for transient cues only (b = −0.014, SE = 0.040, z = −3.36, *p* < 0.001) (see Figure 4).

The difference in strategy use between cue types seems to be driven by a substantial number of participants in the transient cues condition that always copied the information trial exactly, even if it contained no rewarded squares. Fifteen percent of participants taking part with transient cues copied the information trial exactly (blanket copying), in at least 50% of trials that showed fewer than 5 rewards (blanket copying when shown 5 rewards is the correct strategy). In the visible cues condition this figure is only 4.1% of participants. This may indicate that some participants were treating the grid search element of the task, rather than solely the working memory task, as a memory test (although see the discussion for more information).

The reverse strategy, avoiding the squares revealed in the information trial entirely, even when they showed multiple (or even all) of the rewards for that trial, was also used but by a much smaller subset of participants; 1% of participants in the transient cues condition and 5% of participants in the visible cues condition avoided the information cues entirely on at least 50% of trials that showed 1 or more rewards (avoiding the cues entirely when shown 0 rewards is the correct strategy).

#### 3.2.3. Analysis of Errors Made

The type of errors made in different conditions was analysed used a Poisson linear mixed effects model with fixed effects of error type (omission errors, when not repeating rewarded selections, or commission errors of repeating non-scoring squares), block condition and cue type and their interactions. Participant ID was included as a random effect. There were significantly more errors made in the WM condition compared to the control condition (b = 0.313, SE = 0.095, z = 3.31, *p* < 0.001). There was also a significant interaction between error type and block condition (b = −0.783, SE = 0.145, z = −5.40, *p* < 0.001). This was due to more omission errors in the WM blocks compared to the control blocks, and more commission errors in the control blocks compared to the WM blocks (see Table 2). There were also significantly more commission errors when cues were transient compared to when cues were visible (b = 1.65, SE = 0.127, z = 13.1, *p* < 0.001). Finally, there was a significant three-way interaction between error type, block condition and cue type (b = 0.438, SE = 0.180, z = 2.43, *p* = 0.015). Post hoc analysis of the interaction using the emtrends function indicates that when the cues remained visible there were differences in error types between blocks: participants made significantly more omission than commission errors in the WM block (b = 0.671, SE = 0.106, z = 6.34, *p* < 0.001) but there was no difference in error types in the control block (b = −0.113, SE = 0.099, z = −1.14, *p* = 0.256). When cues were transient both blocks had significantly higher rates of commission compared to omission errors (b ≤ −1.42, SE ≤ 0.079, z ≤ −19.7, *p* < 0.001). Mean total errors by participant are shown in Figure 5.

#### 3.2.4. Clustering of Selections

A measure of selection distance was calculated for each set of grid selections (the five grid squares displayed in the information trial, or selected by the participant, on each trial). This took pairwise distances between each square selected in the grid on any particular trial and summed each pair to give a total value. This total value is the selection distance for each set of selections. Larger total values indicate the selections are more spaced out, and smaller values indicate the selections are more clustered. Figure 6 shows the mean selection distance for each condition at each information trial reward level.

Selections made by participants were significantly more clustered than the grid squares displayed in the information trials, as shown by a repeated measures ANOVA (information trial: 53.1, participant selections: 49.1; (F(389) = 247, *p* < 0.001)). The grid squares displayed in each information trial were always selected at random, so more clustering in the participant selections indicates participants were selecting grid squares that were closer together than would be expected by chance. The selection distance for each set of participant selections was analysed using a linear mixed effects model with fixed effects of block condition, cue type, rewards in the information trial, the grid distance of the information trial squares and the interactions between block condition, cue type and rewards. Participant ID was included as a random effect. Selections were significantly more clustered in the WM block (b = −1.53, SE = 0.481, t (8995) = −3.18, *p* = 0.001). Participant selections were significantly less clustered as the number of rewards in the information trial increased (b = 2.11, SE = 0.111, t (8995) = 18.9, *p* < 0.001), and when the information trial was less clustered (b = 0.377, SE = 0.011, t (8995) = 34.3, *p* < 0.001). The difference between WM and control selections was greater when cues were visible (b = 1.41, SE = 0.670, t (8995) = 2.10, *p* = 0.035), although when the number of rewards in the information trial was high, the difference was greater when cues were transient (b = −0.445, SE = 0.220, t (8995) = −2.03, *p* = 0.043) (see Figure 6 for both interactions). Cue type did not have a significant effect on selection distance (b = 1.30, SE = 0.773, t (8995) = 1.69, *p* = 0.092).

#### 3.2.5. Simulated Transmission Chain

We simulated whether agents that behaved in the same way as real participants would show a ratchet effect in a transmission chain. This would indicate whether the experimental manipulations used in the task prevented cumulative improvement of scores over multiple generations. The simulation was run in PsychoPy 3.0.3 [23]. Code to run the simulation is provided via the OSF (see Data Availability Statement).

At generation 1 of the simulation, an agent would receive a score based on the score of a randomly selected real participant from a trial that was shown 0 rewards. That simulated score would then form the input for an agent in generation 2. For example, if the score at generation 1 was 0.5 points due to 1 reward being found in the grid, the score of the agent at generation 2 would be drawn from a real participant on a trial where the information trial showed 1 rewarded square. The simulation ran for 40 generations and was repeated 5000 times. Figure 7 shows the average score at each generation across all runs of the simulation. Optimal behaviour (always repeat rewarded squares and never repeat unrewarded squares) and random behaviour (select five random grid squares on every trial) are also shown on the graph.

The simulation shows that even though the task was simple, it was not trivial. Optimal behaviour required around 40 generations of participants to reach the maximum possible score; random behaviour never found on average more than 1 rewarded square per trial. Agents did not perform optimally in any condition, and differences between the conditions are minimal. However, agents behaving as if they were in the control blocks did reach a higher maximum score, and reached this marginally sooner, than agents with reduced working memory access.

#### 3.2.6. Metacognition Task

Metacognitive accuracy was around 75% (Control block: 75.5%, WM block: 73.1%), although there was substantial individual participant variation in discrimination sensitivity (control block range: 62.5–91.7%, WM block range: 54.2–91.7%). Data from one participant were removed from the analysis due to floor effects, as their performance in both blocks was below 60% accuracy, which was not significantly above chance in either block (as shown by a binomial test: *p* ≥ 0.541). Eight participants scored below 60% accuracy in the WM block. Two participants scored above 90% in the WM block and two scored above 90% in the control block. However, none of these participants were removed from the analysis as the floor and ceiling effects were not consistent across blocks. Their unusually high or low performance in certain blocks may therefore have reflected real effects of the experimental manipulation. A repeated measures ANOVA showed no significant difference in discrimination accuracy between control and WM blocks (F(49) = 2.0, *p* = 0.164). Participants in both blocks tended to rate their confidence highly, with much higher rates of responding at the higher end of the confidence scale (see Figure 8).

Success on each trial was analysed using a binomial general linear mixed effects model with fixed effects of block condition, confidence, target side (whether the correct stimulus to pick was on the left or right) and the interaction between block condition and confidence. Participant ID was included as a random variable and the control block was taken as the baseline. The model was significantly better than the null equivalent (χ^2^ (4) = 43.9, *p* < 0.001). Participants were significantly more successful when they were more confident (b = 0.834, SE = 0.251, z = 3.312, *p* < 0.001), although this result may be affected by the very low number of responses given at the lower end of the confidence scale (see Figure 8). There was also a significant effect of target side (b = 0.409, SE = 0.096, z = 4.24, *p* < 0.001), with a higher chance of success on each trial if the target stimulus was on the right. This shows a strong right-side bias in responses, despite an approximately equal balance of the left and right stimuli being correct.

The impact of the dual-task on participants’ metacognition was analysed using the metaSDT package in R [28]. A score of metacognitive efficiency (m-ratio) for each participant in each block was calculated as metacognitive sensitivity (meta-d’_b_) divided by overall sensitivity in the visual task (d’). Meta-d’ is defined as the type-I sensitivity (accuracy in the visual discrimination task) that would be found if all of a participant’s type-II ratings (confidence ratings) were considered to be optimal [29]. This measure aims to give a bias free measure of metacognition which is not affected by performance in the visual task [21]. Metacognitive efficiency, as measured by m-ratio, was fairly low, around 50% for both blocks, and a repeated measures ANOVA showed no significant difference between control and WM blocks (mean control: 0.544, mean WM: 0.406, F(49) = 0.174, *p* = 0.679).

#### 3.2.7. Concurrent Distractor Tasks

For the grid search main task, accuracy in the distractor task was almost at ceiling in the control block for both cue conditions, and high but not close to ceiling in both cue conditions of the WM block (see Table 3).

A binomial linear mixed effects model with fixed effects of block condition, cue type and their interaction, and participant ID included as a random effect, found that accuracy was significantly lower in the WM block compared to the control block (b = −2.01, SE = 0.130, z = −15.4, *p* < 0.001). There were no significant effects of cue type or the interaction between block condition and cue type (b ≤ |0.163|, SE ≤ 0.188, z ≤ |0.869|, *p* ≥ 0.385). This model was significantly better than the null equivalent (χ^2^(3) = 769, *p* < 0.001).

In the metacognition task, accuracy in the distractor task was also at ceiling in the control block and high in the WM block (see Table 4).

A binomial linear mixed effects model with a fixed effect of block condition and participant ID included as a random effect, found that accuracy was significantly lower in the WM block compared to the control block (b = −2.55, SE = 0.223, z = −11.4, *p* < 0.001). This model was significantly better than the null equivalent (χ^2^(3) = 220, *p* < 0.001).

As the concurrent task accuracy was significantly lower in the WM block, additional analyses were run on a subset of the data including only trials in which the distractor task was answered correctly. In these trials, participants were more likely to be actually using their working memory to answer the question, rather than focusing solely on the main task. These analyses (presented in full in the Appendix A) showed the same main effects and interactions as the original data set. Therefore, any impact from offloading of main task demands to the concurrent working memory task was considered to be minimal.

## 4. Discussion

It should be noted that, as the EMCC has not been empirically tested before, the studies outlined above aimed primarily to establish the feasibility of a method for testing the EMCC, more so than to provide a conclusive answer regarding the validity or otherwise of this theory. In the discussion points below, therefore, we are not trying to make definitive claims about the nature of cumulative cultural evolution. Instead, we aim to assess whether the EMCC remains a plausible hypothesis for future consideration.

### 4.1. Grid Search Task and Simulation

Participants were able to score significantly higher on the grid search task when the information trial contained more rewarded squares (and therefore more bits of information) and scored lower overall in the transient cues condition. This difference was not affected by the distractor task. This showed that adults were able to demonstrate a ‘potential for ratcheting’ (PFR) [15], even under additional memory load.

The observed decrease in outperformance with an increase in information trial rewards is to a certain extent expected, as the higher the number of rewards in the hint, the fewer rewards remain in the grid. However, the rate of decline was significantly steeper when under additional working memory load. This is reflected in the interaction analysis of correct strategy use; strategy use is better in the WM block when the information trial contains few rewards but worse when rewards are high (discussed further below). Additionally, significantly more errors were made in the WM blocks and when cues were transient—both conditions that imposed additional demands on participants’ working memory. This may be due to higher memory requirements when the information trial had a higher number of rewards. Taken together, these findings suggest some working memory involvement in individuals displaying high levels of PFR. This suggests some support for the EMCC, as it implicates memory requirements (and therefore system-2 involvement) in the capacity to consistently outperform demonstrated information.

The level of underperformance at high information trial reward levels revealed substantial errors across all conditions. However, the analysis of error types suggests that additional working memory loads affected the type of errors participants made. In the WM blocks, a significant majority of errors were *omission* errors, whereas in the control blocks it was the opposite. This may indicate that the benefit of working memory access for PFR is in preventing omission errors, rather than guiding efficient exploration of unknown space. The errors analysis, therefore, is not consistent with the idea that system-2 processes are facilitating selective copying. Rather, the analysis suggests there may be some benefit of working memory access for high-fidelity copying. There is therefore mixed support for the EMCC. Although the specific predictions made in the introduction are not met, system-2 involvement, that relies on high memory capacity, may still be beneficial to CCE even if the benefits are not due to an increased propensity for selective copying.

The significant effect indicating a higher number of errors in the WM block appears to be at odds with the higher levels of optimal strategy use seen in the WM block. This apparent paradox is caused by the significant interaction between block condition and error type and is driven largely by a small number of participants who made substantially more of one type of error depending on the block condition. However, these participants were not treated as outliers (and thus not removed from the data) for two reasons. Firstly, the outlying behaviour seems to be caused by the experimental manipulation between blocks. Secondly, many participants made a similar number of errors under different cue conditions, so the large number of errors made by some participants present as outliers only when the data are split by both error type and cue type.

There was a significant difference in correct strategy use between cue conditions. When cues were visible (GSV) there was a consistently high use of the correct strategy in both control and WM blocks. However, when cues were transient (GST), correct strategy use in both blocks increased significantly with the number of rewards in the information trial. This difference was largely driven by a sub-group of participants in GST that frequently copied the entire information trial, rather than exploring other grid locations, even when shown few or no rewarded squares. This strategy use is unlikely to be explained by a poor understanding of the task instructions, as mean accuracy in the understanding checks was 3.77/4. A practise trial was also included in both blocks. If participants re-selected unrewarded squares during this practice they were prompted again that they could search the whole grid. In addition, the task instructions were updated between public and lab testing to make it more unambiguous that the entire grid should be searched for the rewards, but the blanket copying strategy persisted in the lab.

Another explanation for the different strategy use may be the more challenging nature of the task in the transient cues condition. The additional cognitive load of remembering the locations of the information trial squares, as well as their reward value, may have been more challenging for some participants. They may have therefore employed a strategy of “repeat all observed squares” to ensure that any known rewards were always repeated, at the expense of missing out on further exploration. This is somewhat similar to the search strategy employed by the adult participants tested by Schulz, Wu, Ruggeri, and Meder [30], who had a tendency to exploit known rewards rather than explore unknown space. Support for this explanation may be found in the over-imitation literature. Schleihauf and Hoehl [31] propose that the use of system-1 processing may lead to blanket copying, via a learned heuristic to save cognitive resources. Although the blanket copying strategy in these tasks was used when the working memory load was both present and absent, it may be the case that the additional memory load introduced by the transient cues was sufficient to trigger system-1 processing. This would indicate strong involvement of memory resources in efficient selective copying.

The 5 s time limit for each trial, although being sufficient to make all five selections with some deliberation, may also have caused participants to use their system-1 for all trials, even in the control block. This could explain the lack of differences between blocks, and could also be a cause of the sub-optimal ratcheting observed across conditions in the simulation. If the trial time limit explains some of the gaps between optimal behaviour and the control blocks in the simulation, it could indicate a substantial role for system-2 processes, if not specifically working memory, in cumulative ratcheting over generations. If so, this might offer more robust support for the explicit, although not necessarily the metacognitive, element of the EMCC. Indeed, simulated multi-generational transmission of a similar search task with no time limit restriction showed that simulated populations of participants rapidly reached the maximum possible score [32]. Further study using this paradigm with the time limits removed could potentially establish if this interpretation is substantiated (being mindful of the reasons time limits were initially included—see *procedure* above).

Overall, the simulation showed that the behaviour of participants that show a potential for ratcheting does lead to an increase in scores when that behavioural data are used to simulate multi-generational transmission. This confirms the validity of the PFR task in testing for the capacity for cumulative culture, especially as the simulation of optimal responding produces sufficient score accumulation that the maximum available score is reached. However, the similarity between all four experimental conditions indicates that, contrary to our predictions, restricted access to working memory does not necessarily prevent ratcheting across generations.

The simulation also illustrates the likely population-level impact of participants’ failure to act optimally, across all conditions. This was partly due to the sub-optimal copying strategy used by a small number of people in the visible cues condition (GSV) and a large number of people in the transient cues condition (GST). There were also a small number of participants who shifted away from rewarded squares shown to them in the hint, predominantly in GSV. The difference between the type of sub-optimal strategy used in the different cue conditions may explain the similarity between the outcome of the simulation for both conditions: score improvement may have been slowed by redundant repeating of non-scoring squares (commission errors) in GST, whereas it may have increased faster but showed more score decreases due to omission errors in GSV. The presence of even a very few shifts away from rewarded squares, coupled with the difficulty in finding all 5 rewards within the grid, prevents the overall population from reaching the maximum score.

It is worth drawing attention to the contrast between our finding and those from other studies of cultural evolution that have utilised task paradigms which examine copying specifically (i.e., tasks in which the optimal response is always to repeat). Studies using copying tasks have often shown steep increases in scores over generations due to participants’ tendencies to form clusters of selections. This is noticeably prevalent in participants that have limited (due to age [33] or species [34]) or restricted (due to processing bottlenecks [35]) working memory capacity. Clustering allows for more faithful replication by subsequent generations due to the increased regularity of the search space, so the task becomes easier over generations. The tendency to shape the input to be more easily learnable is a feature of many studies of cultural evolution, particularly in tasks which have conventional (i.e., copy this image) goals, and likely reflects weak learning biases of the participants [36]. The present study, in contrast, has an instrumental (i.e., score as many points as possible) goal and the form of the information trial was never shaped by a previous participant to become more easily copied. In fact, as the information trials become more high scoring, finding the remaining rewarded squares becomes more challenging as the size of the search space remains the same but the number of rewards to be found decreases. This is therefore more analogous to many examples of human cumulative culture in the ‘real world’; the more accumulated knowledge required to form a trait, the more difficult it is to reproduce faithfully and the harder it is to improve upon [37]. This pattern is observed in miniature in the results presented above. In the grid search task, participants struggle to match the information trial when it scores more highly. Similarly, even optimal performance in the simulation shows rapid score accumulation while there are still multiple rewards to be found in the grid which slows as fewer rewards remain. This again highlights the value of using this method to assess capacities for CCE under different experimental manipulations.

Some clustering was observed in the grid search task. Participants made selections that were more clustered than the random selections displayed in the information trials and made more clustered selections in the WM blocks than in the control blocks. This indicates that, as in a task with a goal of copying, participants have a weak bias to cluster their responses. The increased clustering in the WM blocks therefore suggests that, in line with the predictions of the EMCC, restricted memory access causes deficits in accumulation of benefits over generations. This deficit may be due to the restriction of explicit processes leading to greater reliance on system-1 biases, in turn leading to reduced ratcheting over generations.

### 4.2. Metacognition Task

Participants’ accuracy in the metacognition task was higher when they reported higher confidence in their responses, showing they did have some metacognitive awareness. However, average metacognitive efficiency across conditions was fairly low. There was also no significant difference between experimental conditions, indicating that increased demands on working memory resources did not affect efficient metacognitive responding.

To establish whether the outcome of the metacognition task supports the EMCC, it should be considered in conjunction with the results of the PFR task and simulation. Contrasting results across the two main tasks provide little in the way of support for the EMCC, as our findings suggest that metacognition was not affected by the same processes as participants’ potential for ratcheting. One conclusion drawn from the PFR task may be that working memory is beneficial for high-fidelity copying, which in turn is beneficial for cumulative cultural evolution. The finding of no deficit in metacognitive monitoring with reduced memory capacity, however, indicates little support for the metacognitive aspect of the EMCC as metacognitive efficiency was unaffected by the dual-task manipulation. As the simulation showed that some ratcheting can occur even with restricted working memory access, the finding that metacognitive monitoring accuracy was not inhibited by the memory load does not rule out the role of metacognition in facilitating that ratcheting. However, it still offers limited support for the EMCC as it shows no evidence that metacognition and ratcheting are disrupted by the same task manipulations.

A final interpretation is that the PFR results were influenced by the time constraints on each trial of the main task. Time limits may have prompted participants to use their system-1 processing throughout, which may explain the sub-optimal responding from participants even in the control block. As the same time limit was also used during the metacognition main task, this interpretation may explain why metacognitive efficiency was similarly low in both testing blocks. A replication of the metacognition task with no time limits, or extended time limits, could provide more evidence to support or refute the EMCC. If restriction of explicit processes throughout the experiment was reducing differences between the control and WM blocks, this may indicate a substantial role for explicit processes in ratcheting over generations and metacognitive monitoring, even if working memory specifically is not implicated.

Two additional considerations should also be explored when interpreting the metacognition task data. Firstly, the outcome of the present study might have been affected by the strong biases towards responding at higher confidence levels and responding to the right-side stimulus. The side bias was so strong in some participants it appears that they may have been responding on the right simply because they used their right hand to respond and this biased them towards touching the right hand side of the screen. This may have additionally influenced the bias towards high confidence levels as always selecting the right-side edge of the screen means selecting the top end of the right-hand confidence scale. A small number of participants had a bias towards selecting the left side, which could also result in higher confidence values given for the left stimulus simply as a result of selecting more often on the left edge of the screen. This combination of side biases may reflect the handedness of participants in the sample, but this cannot be confirmed as handedness was not collected in the participant information. Although the meta-d’ method of assessing metacognitive efficiency is designed to be robust against type-I and type-II response bias, the measure can be affected by very high or low values for both type-I and type-II hit and false alarm rates [28]. With such a strong bias towards one response side and the high end of the confidence scale, the type-II false alarm rate for some participants would be very high. This may then indicate that the lack of a difference between conditions is exacerbated by meta-d’ measures that have been influenced by extreme inequality in the responses. This potential limitation could be combatted in future testing by changing the orientation of the stimuli to make it less likely that participants would always respond with the same part of the scale.

Secondly, the metacognition task investigated only metacognitive monitoring accuracy. It may be the case that the experimental manipulations had an impact on metacognitive *control* processes that were not detected, leading to an underestimation of dual-task impact on metacognition. It is easily conceivable that while metacognitive monitoring is beneficial in assessing the level of one’s own knowledge, it is metacognitive control that enables the use of the selective copying strategies that are argued to be a requisite for CCE. This interpretation would therefore suggest that future testing paradigms should examine the role of both metacognitive monitoring and control on potential for ratcheting.

## 5. Conclusions

This study aimed to establish whether having reduced access to explicit, system-2 processes negatively impacted capacities for cumulative cultural evolution in humans. We experimentally tested the EMCC (the hypothesis that system-2 executive resources and explicit metacognition are required for cumulative cultural evolution to occur). This was carried out by manipulating access to working memory (in order to manipulate access to system-2 processes) while participants’ *potential for ratcheting* was tested, via use of a dual-task paradigm. The results indicate partial support for the EMCC: there is some evidence of explicit processes facilitating efficient copying, and the restriction of explicit resources leading to reduced capacity to perform optimally in a grid search PFR task. No clear evidence of metacognition being impacted by the same task manipulations was found.

The impact of reduced working memory resources on metacognitive monitoring appeared to be minimal. This is in contrast to previous findings (e.g., [38]), but may have been distorted by short time constraints within the task or strong response-side biases in responding. Differences in metacognitive control may also have occurred but not been detected by the task. This finding can therefore neither conclusively support nor refute the metacognitive element of the EMCC.

In the grid search task, there was some evidence for decreased working memory resources leading to a reduced ability to copy selectively and efficiently. When more copying and less exploration was required, the use of optimal search strategies was generally lower in the WM block compared to the control block. Additionally, participants made more errors in the WM block, particularly errors in accurate copying of known rewards. Participants in the transient cues condition may also have been using a learned system-1 heuristic to search the grid, due to the additional working memory load imposed by the disappearing cues. Finally, participants were more likely to cluster their responses in the WM blocks compared to the control blocks, which may indicate increased reliance on an implicit bias. These findings suggest that selective social learning strategies may be more efficient if the copier has full access to their working memory resources.

When iterated over multiple generations, differences between the blocks and conditions were no longer apparent. Similar levels of ratcheting were seen with both full working memory access and substantial working memory loads, although ratcheting was also sub-optimal in all experimental conditions. The results presented above therefore indicate that, although beneficial for efficient copying, abundant working memory resources are not a necessary requirement for cumulative score improvement over generations However, short time constraints on each trial of the grid search task may have impacted participants’ access to explicit processes, even in the control block, contributing to an artificially large gap between the control block and optimal responding, and possibly supporting the idea of a far more substantial role for explicit processes in ratcheting potential.

Worthwhile directions for future research on this topic might include direct investigation of the effect of varying time constraints on each trial, and comparisons between metacognitive control and monitoring processes.

## Figures and Tables

**Figure 1 entropy-24-00325-f001:**
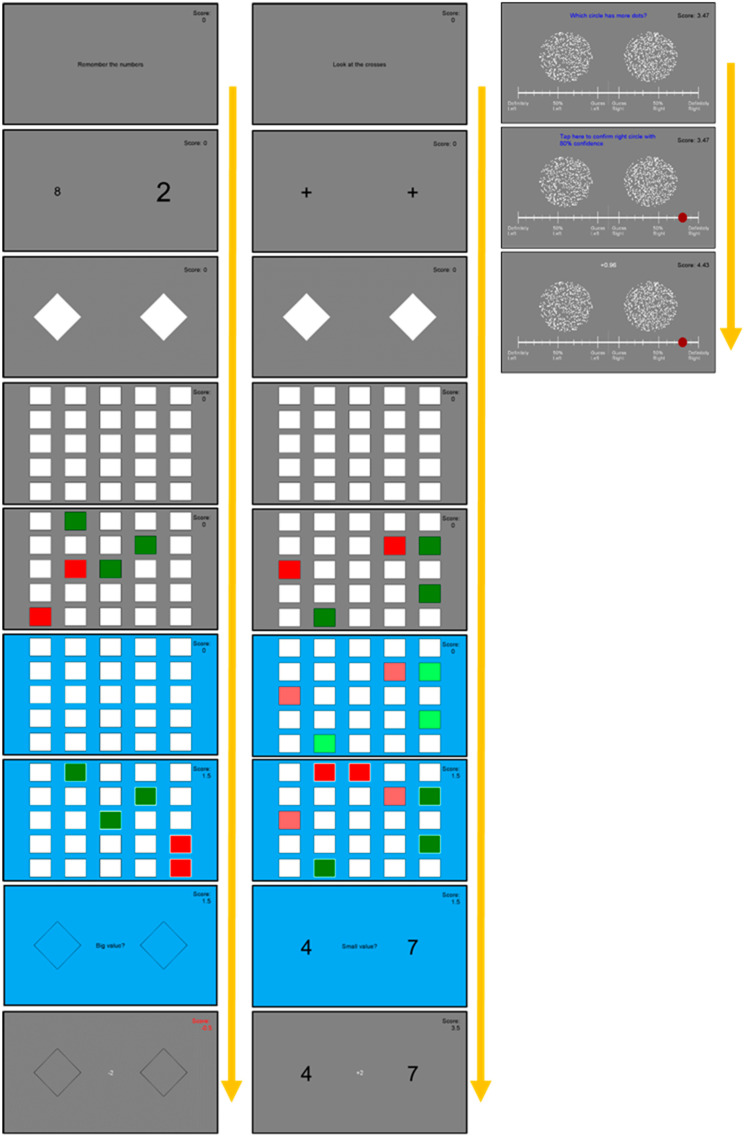
(**Left**): An example of 1 full trial of the grid search task in the transient cues condition, completed with the executive function concurrent task. (**Middle**): An example of 1 full trial of the grid search task in the visible cues condition, completed with the control concurrent task. (**Right**): An example of 1 trial of the metacognition visual discrimination task, main task only. Participants were instructed to only touch the screen when either the background was blue (grid search task) or the text was blue (metacognition task) to avoid premature touches being incorrectly counted.

**Figure 2 entropy-24-00325-f002:**
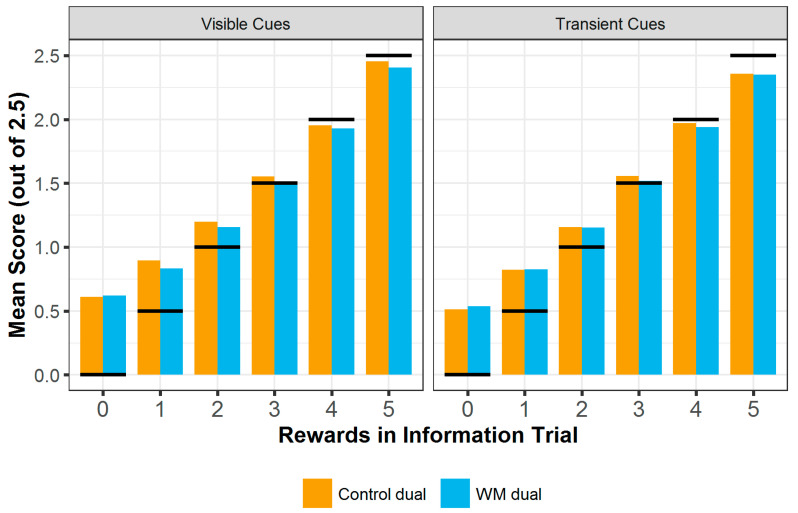
Mean score gained at each number of rewards shown in the information trial hint, split by block condition and cue type. Black horizontal lines indicate the information trial score at each reward level. This is the score that would be required to exceed in order to outperform the information trial. Across conditions, participants easily outperform the information trial when the information trial shows few rewards, but struggle to even match performance (see also Figure 3) when the information trial is high scoring.

**Figure 3 entropy-24-00325-f003:**
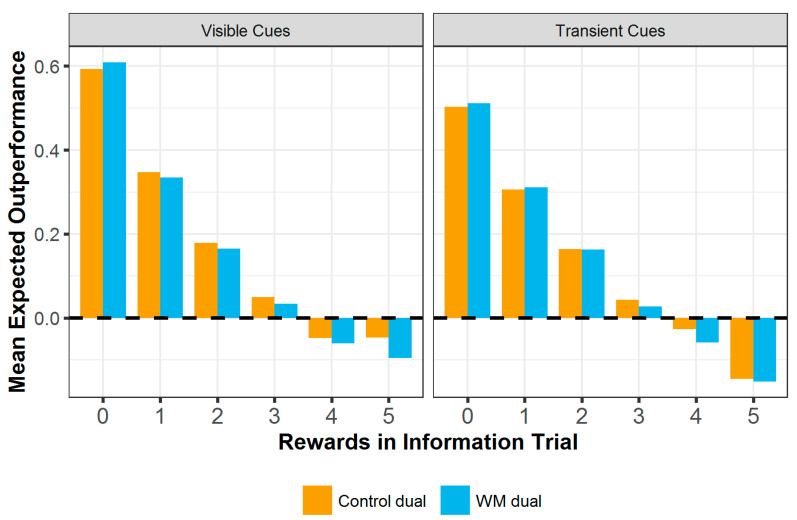
Mean outperformance of the information trial at each hint level. The dashed line indicates performance required to match the score of the hint. Bars below this line indicate mean performance was lower than the information trial. Based on the strategies used, participants in both blocks can outperform the information trial when it shows few rewards, but struggle to even match performance when the information trial is high scoring.

**Figure 4 entropy-24-00325-f004:**
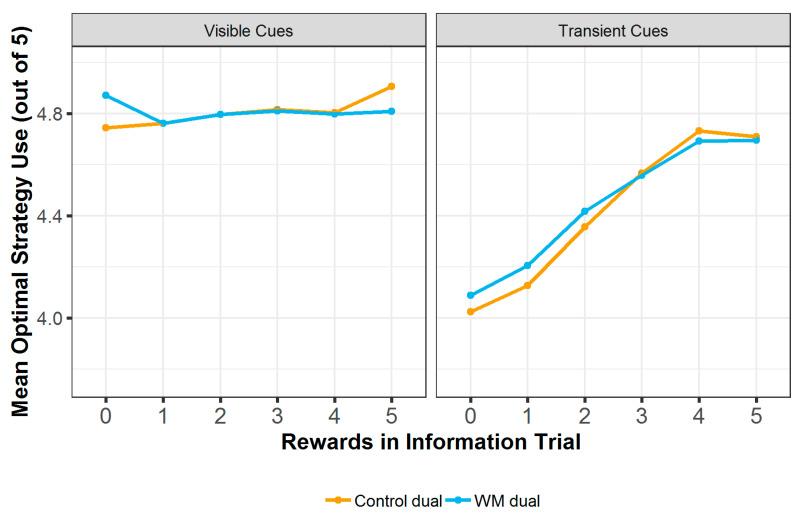
Optimal strategy use at each level of rewards shown in the information trial, split by block condition and cue condition.

**Figure 5 entropy-24-00325-f005:**
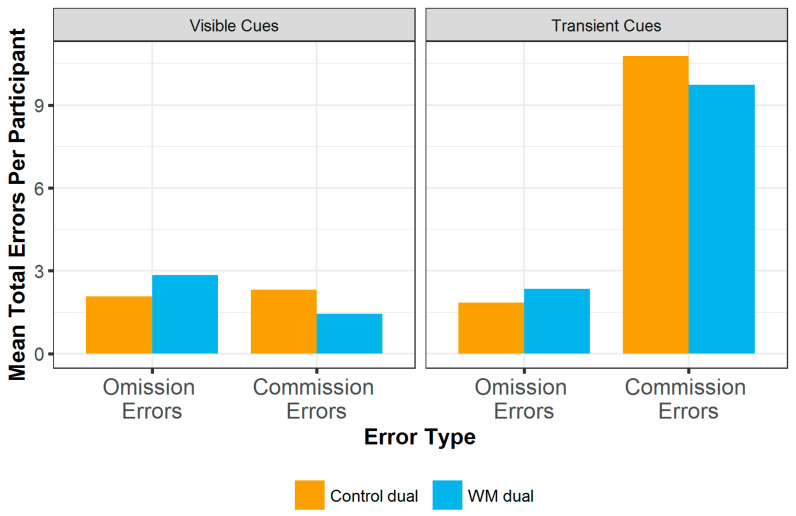
Mean total errors per participant, per block. Significantly more errors were made in the WM block. When cues are visible participants make more omission errors compared to commission errors in the WM block, but there is no difference between error types in the control block. When cues are transient, participants make more commission errors compared to omission errors in both blocks.

**Figure 6 entropy-24-00325-f006:**
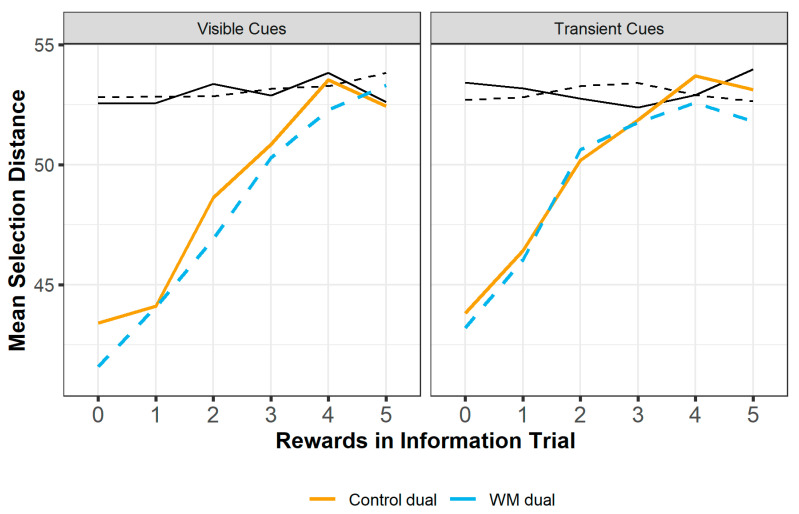
Mean selection distance at each reward level, split by block condition and cue type. Black lines indicate the selections displayed in the information trial. Solid lines are the control block, dashed lines are the WM block. Participants made selections that were more clustered (lower selection distance) than the information trial and participant selections were more clustered in the WM block than the control block. The more rewards shown in the information trial, the more the participant selections resembled the clustering of the information trial.

**Figure 7 entropy-24-00325-f007:**
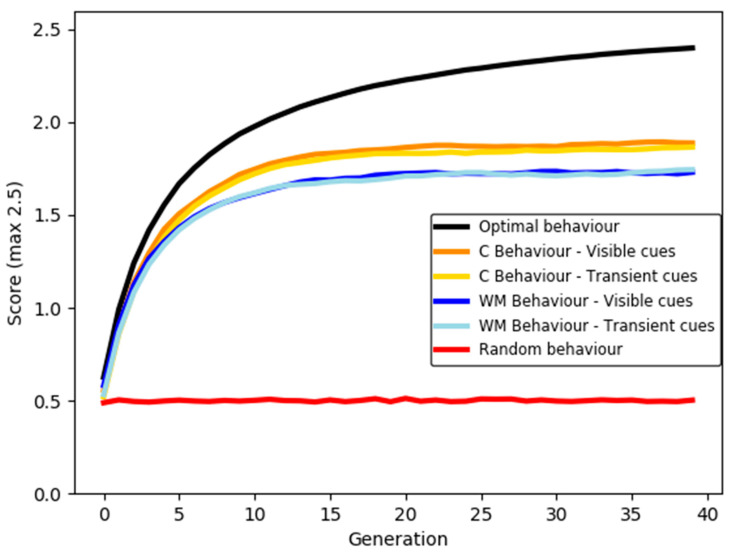
Simulated scores at each generation of a transmission chain created using real participant scores for each condition, compared with optimal and random behaviour.

**Figure 8 entropy-24-00325-f008:**
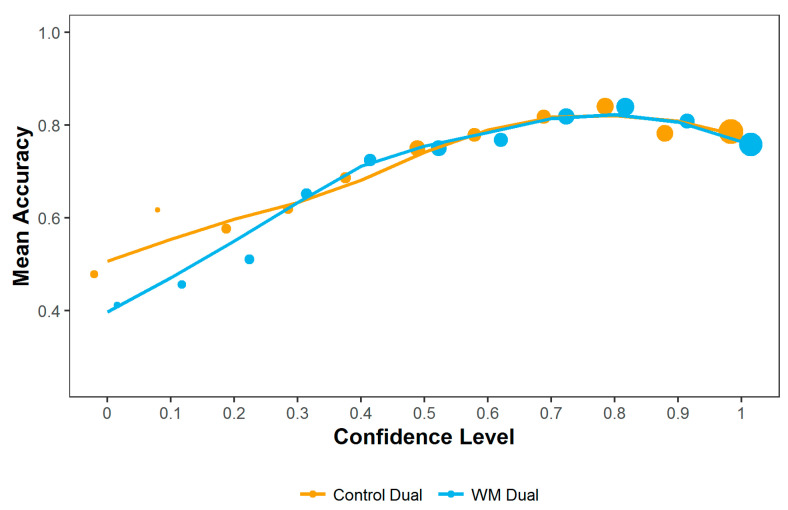
Mean accuracy of patch discrimination at each confidence level. Size of points indicates the number of responses given at that confidence level, with larger dots indicating more overall responses.

**Table 1 entropy-24-00325-t001:** Distribution of participants in different tasks, conditions and testing locations.

Task	Centre for Life Sample	Lab Sample	Total Sample
Grid Search—Transient Cues	55	44	99
Grid Search—Visible Cues	53	44	97
Metacognition	51	0	51
Total	159	88	247

**Table 2 entropy-24-00325-t002:** Mean total errors by participant, split by cue condition, block condition and error type.

Cue Condition	Block Condition	Mean Omission Errors (Total Per Participant)	Mean Commission Errors (Total Per Participant)
**Visible Cues**	**Control**	2.08	2.32
**Visible Cues**	**WM**	2.84	1.45
**Transient Cues**	**Control**	1.84	10.8
**Transient Cues**	**WM**	2.40	9.74

**Table 3 entropy-24-00325-t003:** Mean accuracy in the concurrent working memory task when completed alongside the grid search task.

Block	Cue Type	Mean Distractor Task Accuracy
**Control**	**Visible Cues**	96.6
**Control**	**Transient Cues**	95.8
**WM**	**Visible Cues**	80.3
**WM**	**Transient Cues**	76.0

**Table 4 entropy-24-00325-t004:** Mean accuracy in the concurrent working memory task when completed alongside the metacognition task.

Block	Mean Distractor Task Accuracy
**Control**	97.9
**WM**	80.1

## Data Availability

All data and analysis code, as well as scripts to run the experimental tasks are available at: https://osf.io/dhvmy/?view_only=b55fbf7fbe414fb9a40c13bb6a213dc6 (accessed on 5 December 2021).

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
