# Peer review of "Restricted Access to Working Memory Does Not Prevent Cumulative Score Improvement in a Cultural Evolution Task"

_entropy, 2022, doi:10.3390/e24030325_

Round 1

Reviewer 1 Report

The article entitled “restricted access to working memory does not prevent cumulative score improvement in a cultural evolution task” discusses the role of working memory in the emergence of cumulative evolution. The key idea of the study is to use a dual task to impact working memory during either a grid-search task (the key task here to assess cumulative effects or “potential for ratcheting” effects) or a metacognitive task. The findings suggest that taxing working memory resources does not prevent the participants from showing cumulative effects.

The question addressed is very timely and the experimental design is clever. The authors report a series of very interesting results, which could make the paper a very nice contribution to the topic. I have only three main concerns, two about the framing of the paper, and one about the ecological validity of the task used.

Point 1.

The key message of the paper is not clear. The title focuses on “working memory” and the term metacognition is not mentioned. However, very soon in the abstract as well as in the introduction the discussion turns on metacognitive skills, and we understand that the use of the dual task is not to assess “directly” the role of working memory but rather to do an experimental manipulation that targets metacognitive skills. The rationale is that the more the working memory resources are taxed, the less the participants can use metacognitive strategies. Nevertheless, later in the discussion, the authors come back on the other potential effects of working memory on the cumulative effects, so that the reader is a little bit lost: Is it a paper on the role of working memory on cumulative culture or of metacognitive skills? I think that the authors should reframe the paper to make all these aspects more explicit because it is hard to understand what the key message of the paper is.

Point 2.

The introduction but above all the discussion are very long. They must be shortened considerably. The structuration of the different paragraphs is also hard to follow. Sometimes, there is also one sentence for a paragraph (ex: lines 75-77). This is very difficult to grasp the macro-structure of the text because of the presence of many very short paragraphs. The discussion section on grid search task and simulation is too long and could benefit from summarizing some points and avoid repetitions.

Point 3.

I wonder whether the authors could elaborate a little bit more on the “ecological” validity of their task. Can we consider that grid-search task is comparable to learning how to make a tool, for instance? I do not criticize the use of lab tasks for exploring cumulative evolution. I think that the task used here is interesting and can provide interesting insights into how cumulative evolution works. However, do the authors think that this grid-search task captures a very specific aspects of cumulative evolution or can be generalized to any domain of cumulative evolution?

Author Response

I would like to thank the reviewer for their kind and helpful feedback on this manuscript, which has given me the opportunity to improve the paper and make it more accessible to readers. Responses to each point in turn are given below, in blue text.

The article entitled “restricted access to working memory does not prevent cumulative score improvement in a cultural evolution task” discusses the role of working memory in the emergence of cumulative evolution. The key idea of the study is to use a dual task to impact working memory during either a grid-search task (the key task here to assess cumulative effects or “potential for ratcheting” effects) or a metacognitive task. The findings suggest that taxing working memory resources does not prevent the participants from showing cumulative effects.

The question addressed is very timely and the experimental design is clever. The authors report a series of very interesting results, which could make the paper a very nice contribution to the topic. I have only three main concerns, two about the framing of the paper, and one about the ecological validity of the task used.

Point 1.

The key message of the paper is not clear. The title focuses on “working memory” and the term metacognition is not mentioned. However, very soon in the abstract as well as in the introduction the discussion turns on metacognitive skills, and we understand that the use of the dual task is not to assess “directly” the role of working memory but rather to do an experimental manipulation that targets metacognitive skills. The rationale is that the more the working memory resources are taxed, the less the participants can use metacognitive strategies. Nevertheless, later in the discussion, the authors come back on the other potential effects of working memory on the cumulative effects, so that the reader is a little bit lost: Is it a paper on the role of working memory on cumulative culture or of metacognitive skills? I think that the authors should reframe the paper to make all these aspects more explicit because it is hard to understand what the key message of the paper is.

The focus on working memory is included as it is the use of working memory that is considered the key distinction between system-1 and system-2 processes. This is relevant to the metacognition framing as it is specifically explicit, metacognitive process discussed in the EMCC that we believe to be relevant to cumulative culture. I hope that the more streamlined introduction in the updated manuscript makes this relationship clearer. I have also updated the conclusion to reiterate the links between explicit processes and the use of a working-memory distractor task.

Point 2.

The introduction but above all the discussion are very long. They must be shortened considerably. The structuration of the different paragraphs is also hard to follow. Sometimes, there is also one sentence for a paragraph (ex: lines 75-77). This is very difficult to grasp the macro-structure of the text because of the presence of many very short paragraphs. The discussion section on grid search task and simulation is too long and could benefit from summarizing some points and avoid repetitions.

The manuscript in its initial format was very long. I have therefore taken this opportunity to remove sections or paragraphs that did not add to the overall value or framing of the studies. I have aimed to remove sections specifically from the introduction and the grid-search task section of the discussion.

Point 3.

I wonder whether the authors could elaborate a little bit more on the “ecological” validity of their task. Can we consider that grid-search task is comparable to learning how to make a tool, for instance? I do not criticize the use of lab tasks for exploring cumulative evolution. I think that the task used here is interesting and can provide interesting insights into how cumulative evolution works. However, do the authors think that this grid-search task captures a very specific aspects of cumulative evolution or can be generalized to any domain of cumulative evolution?

I would actually say that the ecological validity of the task is fairly low – due to the nature of dual-task studies needing to be very tightly controlled the main tasks (grid search and metacognition in this instance) need to be fairly simple in order to rule out confounding task interactions and this simplicity does require some compromise with ecological validity or generalisability to other cultural domains. The value in very tightly controlled tasks such as this is that we can test very specific elements of a hypothesis – so in the current MS whether restricting access to system-2 impedes the capacity to apply a flexible copying rule - that might not produce clear results or might not be practical to implement if tested with a more complex or ‘realistic’ cultural evolution task. The end of the introduction has been amended to include this clarification.

Reviewer 2 Report

This was in many ways an excellent paper — it is a nicely executed experiment addressing an interesting issue in the cumulative cultural evolution literature. And I very, very much appreciated both the open data and the methodological care, transparency, and rigour shown by the authors in performing the analyses. Nevertheless, I think this work has several substantial limitations, and I’m not sure that they are fixable. I may be wrong, in which case I’d be happy to review a revision, but I’m not sure I am.

The largest worry is that I am not sure these results tell us much about anything other than the task itself (I.e., we can’t extrapolate from it to anything about cumulative cultural evolution). In a nutshell, the main finding is that people under working memory load are *slightly* less accurate at retaining the information in a grid search task, but they still retain enough to lead to accruing knowledge over the course of generations.

But consider two alternate experiments / patterns of results that we might have seen instead. In one, which I’ll call HARD, either the grid task or the distractor task is so difficult that people can’t use or retain the information at all (this would surely happen at some point with a large enough grid or enough numbers they have to remember in the working memory distractor). In the other, which I’ll call EASY, both the grid task and the distractor task are so simple that there would be no effect of working memory load on the amount of information used in the grid task (consider for instance a situation with a grid size of 9 and a WM task where they have to remember only one number). These experiments would produce opposite patterns of results and opposite conclusions about the EMCC hypothesis, even though they vary only in task demands that the EMCC hypothesis makes no claim about being relevant.

Put another way — the EMCC hypothesis holds that it is strategic learning rather than simple imitation that explains cumulative cultural improvement. In the HARD experiment, by the logic in this paper, the conclusion would be that the EMCC is supported because WM disruption removed people’s capacity to use and build on the information provided. By contrast, by the same logic, in the EASY experiment, the conclusion would be that the EMCC is contradicted because people were unaffected by WM disruption. This makes me very uneasy, because I’m 100% sure that I could design either the HARD or the EASY experiment, which means that which conclusion about the EMCC you wanted to support could be entirely determined “in advance” just by deciding how difficult to make the task. 

To expand on this, I think there are several issues here:

(a) A working memory distractor task potentially has more effects than *just* disrupting strategic thinking. By comparison to the control distractor task, it takes up some of the cognitive capacity that would otherwise be available during the grid task; this excess capacity could have been used for strategising, but it also could have been used for just remembering more things or paying more attention during the information trial. This lowered capacity could explain the performance decrement in the WM condition (or the much larger imagined decrement in the HARD experiment) even if strategising itself wasn’t affected at all. This is consistent with the lack of effect of WM load in the metacognitive task.

(b) Perhaps I’m missing something obvious, but the grid task doesn’t seem to capture strategic learning in an interesting sense vis-a-vis cultural evolution. The idea of the EMCC is that strategic learning involves having the metacognition to reflect and realise what information is good (or not), and then only retain and use the good information. But in the grid task, it doesn’t require metacognition to identify which information is good or not — it comes clearly labeled, with green being good and red being bad. An extremely trivial cue-based learner which copied the good and not the bad could achieve optimally on this task. It doesn’t require social learning or having the kind of insight into one’s own mental state that is typically thought of as being metacognitive. As a result, showing impaired performance on the grid task as a result of WM load doesn’t seem to me to necessarily mean anything about metacognition or explicit learning at all, and thus doesn't necessarily say anything about the EMCC.

Like I said, I might be missing something, but this seems to be to be a fundamental issue to the logic of the paper. If I am missing something I am probably not the only one and I think this logic needs to be spelled out much more clearly in the introduction.

I have a few other comments, more minor but still potentially important:

1. All of the trials on the metacognitive task were of the same difficulty level, which seems possibly problematic. Since the difficulty level was constant yet the metacognitive question required people to estimate their accuracy, this may have resulted in a task that was unable to detect individual differences in metacognitive ability. For instance, suppose you had four people: A could estimate their accuracy reliably even when the patches were only 2% different from each other; B down to 6%; C down to 10%; and D down to 14%. Person A and B would look equivalent on your task, and so would C and D (with any variation from trial to trial being due to just noise, because otherwise the trials are no different from each other). If the task lacks this kind of sensitivity, the end result might be that the WM distractor task *did* affect metacognition, but that relatively few people were close enough to their threshold difficulty to show up as different performance: if the WM distractor knocked Person A down to only being able to estimate accurately when patches were 3% different from each other, Person B down to 7%, C down to 11%, and D down to 15%, that would have been a consistent across-the-board effect that you would not have been able to discern at all. I’m not saying this necessarily happened — I think it’s just as likely that the WM distractor didn’t affect metacognitive abilities as measured by the visual task — but my point is that as the experiment is designed, it is impossible to tell whether this happened or not.

2. I’m concerned about how the analysis may be shaped by the people who didn’t follow the instructions (i.e., who repeated the information trial exactly). It seems quite plausible to me that all of the findings on pages 12 through 14 (and even the difference between transient and visible trials in the main analysis corresponding to Figures 2 and 3) may have been driven by those people. They would have made many more commission errors than omission errors (consistent with Figure 5) and could explain the otherwise somewhat strange result that in transient trials, optimal strategy use was higher when more information was provided (Figure 4). I appreciate the authors’ transparency in disclosing these people, and the discussion about why they were not removed, but I don’t think that decision makes sense. One of the reasons that they were not removed is that the “outlying behaviour seems to be caused by the experimental manipulation between blocks.” However, importantly, it may not have been caused by the aspect of the manipulation the experimenters cared about (the memory demands) but instead by something we do not care about (different interpretation of the instructions across blocks). In that case, transparently removing outliers is not only justified but necessary. Secondly, it was stated that all participants achieved 50% or more in the instruction check, but that means that they could have missed 2 out of the 4 questions, which is a lot. Were any of the questions relevant to this particular aspect of the experiment? (i.e., about whether their task was to copy all of the squares exactly or just the green ones?) If not, then overall performance on the instruction check seems fairly irrelevant. If so, then did those participants get that question right? Do the results change if you only include the participants who got that question right? At the very least, I would like to see a further analysis of this, perhaps in supplemental materials, determining whether the results are robust to the exclusion of these participants (or participants who got that question on the instruction check wrong).

3. In general I think it’s pretty dangerous to over-interpret small findings from mixed-effect models with many variables and interactions, especially when some of the variables involved are probably not normally distributed (e.g., commission errors or score/outperformance by reward). For instance, on page 12 we have significant main effects of cue type and reward, but block type matters only in interaction with reward. In a post-hoc analysis this is explained by less decline in outperformance in the control condition with larger rewards, but looking at Figure 3 this is hardly perceptible at all (only for reward 5 in the visible condition). I suspect that the only reason this is significant is that the main effect of cue type is absorbing so much variance. At most, it is a tiny (TINY) effect, certainly not enough for one of the main findings to be that the rate of decline is steeper under WM load. Similarly, the analysis on page 13 that found significantly more errors in the WM condition was surprising, given that the corresponding Figure and Table appear to show nothing of the sort; it makes me worried that the model may be off if the variables were not normal, or at least that this finding (which is much smaller than the effects of cue and error type) is overstated.] This general worry applies to almost all of the analyses in the paper. 

4. This is minor, but I found it to be a little misleading to claim that participants “struggled to even match performance” when the information trial was high scoring. Based on Figure 2 and 3, they matched or outperformed performance at every level except for five, where it is actually impossible to improve and noise can only go in one direction (i.e., worse performance). It is therefore not interesting to me that people didn’t outperform or even match at level 5.

5. How many trials were there in each block per participant? 

I know I have a lot of criticisms but I hope the authors are not too dismayed. I spent a long time on this review — more than I usually do — because I did find this an interesting and provocative paper, well worth the time. So I hope that either the flaws I pointed out are a result of something I missed, and thus fixable, or else — even if not — I hope my comments are helpful in designing a future study that addresses the same questions but does not have those issues. I do think this is interesting work and I applaud the systematic and transparent approach here, and commend the authors for it.

Reviewer 3 Report

First, I'm not the most qualified person to judge the quality of the experimental setup, but as far as I can see it looks very well thought through and it is well presented.

In a sense, the authors report a negative results, which I think is commendable, and I also think it is interesting. They conclude, as I would, and would have suspected, that it is quite challenging to design tests for how executive functions act to produce cumulative culture. This problem is only exacerbated by the fact that the idea of what cumulative culture really is lacking in rigor generally. I would expect that humans are exceptionally plastic and able to compensate for the loss of one capacity with other capacities in ways that complicates most attempts to control by removing an ability. Maybe we also need a better theoretical map of what executive functions might do in detail to find those detailed predictions that can be tested?

I haven't worked myself with experiments, I should add, and in many cases I've found experiments to be quite inconclusive (and oversold in some cases). Still, nobody can claim that it's not important to perform experiments in general, and I think this is a good contribution to an effort that probably needs a lot of work in the future. I mean, the really big problem to me is that cumulative evolution of culture is something that really happens across many generations. What we see as cumulative within one generation in one person (or a small circle of persons) may not even be cumulative in the long run. We risk mixing up processes as different as development and genetics if we look at biological counterparts.

Some brief notes on the framing, which in a way are minor, but I think they are somewhat important since they are about the framing of the work.

On the first page, the authors state that high-fidelity copying has been established in animals. It hasn't. Those are 15 years old references. Quite to the contrary actually. It is highly doubtful whether even Homo copied at all, let alone with high fidelity, until at least 500,000 years or so ago. There is no unequivocal evidence for it. It is easy to mistake what is often termed "emulation" for imitation since the former frequently shows equifinality even if no copying takes place. 

See e.g.:

Tennie, C., Premo, L. S., Braun, D. R., & McPherron, S. P. (2017). Early Stone Tools and Cultural Transmission: Resetting the Null Hypothesis. Current Anthropology, 58(5), 000–000. https://doi.org/10.1086/693846

Tennie, C., Hopper, L. M., & van Schaik, C. P. (2020). On the Origin of Cumulative Culture: Consideration of the Role of Copying in Culture- Dependent Traits and a Reappraisal of the Zone of Latent Solutions Hypothesis. In L. M. Hopper & S. R. Ross (Eds.), Chimpanzees in Context: A Comparative Perspective on Chimpanzee Behavior, Cognition, Conservation, and Welfare. Chicago University Press.

While copying is found in e.g. birds, that is not a comparable capacity. Birds copying tunes is a highly specialized ability, and it is relatively easy to account for the selection pressures that would bring it about and maintain it in so many species. The ability and usefulness of copying actions and ways of thinking in order to copy outcomes is a much more general ability.

For a very nice discussion about the adaptive valley between animal and human learning (and imitation as the basis for cultural inheritance) see:

Shea, N. (2009). Imitation as an inheritance system. Philosophical Transactions of the Royal Society of London. Series B, Biological Sciences, 364(1528), 2429–2443. https://doi.org/10.1098/rstb.2009.0061

Also, about that social learning is found in many animals: That doesn't reflect negatively on the thesis that copying is important for cumulative culture. What it does say is that social learning does not usually lead to copying and something like human culture.

The authors state (page 1) that "brings the notion of these capacities as sufficient requirements for CCE into question". That's easy to agree with but I'd say the idea is more that high-fidelity is necessary than that it is seen as sufficient today.

Most would easily agree that we need, for example, ways of telling what to copy and not (e.g. prestige-bias, adaptive filtering, and so on). In fact, high-fidelity copying is not even sufficient in biology - in practice all sorts of other processes are needed to make it work beyond the simplest imaginable RNA replicators (which happen to just have such other properties). 

So I think that in terms of the framing of the work, the manuscript could gain by looking a bit into the more modern literature on imitation.

Author Response

Thank you for your comments on the manuscript, especially for noting the importance of sharing null results. I have written my responses to each of your comments below (see blue text).

First, I'm not the most qualified person to judge the quality of the experimental setup, but as far as I can see it looks very well thought through and it is well presented.

In a sense, the authors report a negative results, which I think is commendable, and I also think it is interesting. They conclude, as I would, and would have suspected, that it is quite challenging to design tests for how executive functions act to produce cumulative culture. This problem is only exacerbated by the fact that the idea of what cumulative culture really is lacking in rigor generally. I would expect that humans are exceptionally plastic and able to compensate for the loss of one capacity with other capacities in ways that complicates most attempts to control by removing an ability. Maybe we also need a better theoretical map of what executive functions might do in detail to find those detailed predictions that can be tested?

While it is certainly true that clearer definitions of cumulative culture and executive functions would greatly benefit our understanding of this field, they are unfortunately beyond the scope of the current study.

I haven't worked myself with experiments, I should add, and in many cases I've found experiments to be quite inconclusive (and oversold in some cases). Still, nobody can claim that it's not important to perform experiments in general, and I think this is a good contribution to an effort that probably needs a lot of work in the future. I mean, the really big problem to me is that cumulative evolution of culture is something that really happens across many generations. What we see as cumulative within one generation in one person (or a small circle of persons) may not even be cumulative in the long run. We risk mixing up processes as different as development and genetics if we look at biological counterparts.

I agree that an experimental approach like this can’t capture the true breadth of multi-generational turnover. In that regard, the ecological validity of this kind of study is fairly low, but that trade-off allows us to test specific predictions in a tightly controlled study environment. Some additions to the manuscript have been made at the end of the introduction to acknowledge this.

Some brief notes on the framing, which in a way are minor, but I think they are somewhat important since they are about the framing of the work.

On the first page, the authors state that high-fidelity copying has been established in animals. It hasn't. Those are 15 years old references. Quite to the contrary actually. It is highly doubtful whether even Homo copied at all, let alone with high fidelity, until at least 500,000 years or so ago. There is no unequivocal evidence for it. It is easy to mistake what is often termed "emulation" for imitation since the former frequently shows equifinality even if no copying takes place. 

I really appreciate this feedback. While I would love to include a far more nuanced background of the imitation literature, I also wanted to take on board comments made by Reviewer 1 about the length of the manuscript. I have therefore opted to remove the reference to high-fidelity imitation in animals, in order to at least avoid including outdated information in the MS.

Round 2

Reviewer 1 Report

The authors have addressed all my concerns, I have not other comment.

Reviewer 2 Report

Thank you for the reply to my review -- I think we may still differ on the implications of the results and some of my other points, but I appreciate the authors' point of view and if the editor is happy to accept this so am I.

Reviewer 3 Report

I think all looks fine now!